# Assessing the Impact of Electrosuit Therapy on Cerebral Palsy: A Study on the Users’ Satisfaction and Potential Efficacy

**DOI:** 10.3390/brainsci13101491

**Published:** 2023-10-22

**Authors:** David Perpetuini, Emanuele Francesco Russo, Daniela Cardone, Roberta Palmieri, Andrea De Giacomo, Domenico Intiso, Federica Pellicano, Raffaello Pellegrino, Arcangelo Merla, Rocco Salvatore Calabrò, Serena Filoni

**Affiliations:** 1Department of Engineering and Geology, University G. D’Annunzio of Chieti-Pescara, 65127 Pescara, Italy; david.perpetuini@unich.it (D.P.); d.cardone@unich.it (D.C.); arcangelo.merla@unich.it (A.M.); 2Padre Pio Foundation and Rehabilitation Centers, 71013 San Giovanni Rotondo, Italy; emanuele.fr88@gmail.com (E.F.R.); federicapellicano.com@gmail.com (F.P.); 3Translational Biomedicine and Neuroscience Department (DiBraiN), University of Bari “Aldo Moro”, 70124 Bari, Italy; roberta.palmieri87@gmail.com (R.P.); andrea.degiacomo@uniba.it (A.D.G.); 4Unit of Neuro-Rehabilitation, IRCCS “Casa Sollievo della Sofferenza”, 71013 San Giovanni Rotondo, Italy; d.intiso@operapadrepio.it (D.I.); serena.diba@gmail.com (S.F.); 5Department of Scientific Research, Campus Ludes, Off-Campus Semmelweis University, 6912 Lugano, Switzerland; raffaello.pellegrino@uniludes.ch; 6IRCCS Centro Neurolesi “Bonino-Pulejo”, 98124 Messina, Italy

**Keywords:** electrosuit, cerebral palsy, neurorehabilitation

## Abstract

The aim of this study is to evaluate the effectiveness of electrosuit therapy in the clinical treatment of children with Cerebral Palsy, focusing on the effect of the therapy on spasticity and trunk control. Moreover, the compliance of caregivers with respect to the use of the tool was investigated. During the period ranging from 2019 to 2022, a total of 26 children (18 M and 8 F), clinically stable and affected by CP and attending the Neurorehabilitation Unit of the “Padre Pio Foundation and Rehabilitation Centers”, were enrolled in this study. A subset of 12 patients bought or rented the device; thus, they received the administration of the EMS-based therapy for one month, whereas the others received only one-hour training to evaluate the feasibility (by the caregivers) and short-term effects. The Gross Motor Function Classification System was utilized to evaluate gross motor functions and to classify the study sample, while the MAS and the LSS were employed to assess the outcomes of the EMS-based therapy. Moreover, between 80% and 90% of the study sample were satisfied with the safety, ease of use, comfort, adjustment, and after-sales service. Following a single session of electrical stimulation with EMS, patients exhibited a statistically significant enhancement in trunk control. For those who continued this study, the subscale of the QUEST with the best score was adaptability (0.74 ± 0.85), followed by competence (0.67 ± 0.70) and self-esteem (0.59 ± 0.60). This study investigates the impact of the employment of the EMS on CP children’s ability to maintain trunk control. Specifically, after undergoing a single EMS session, LSS showed a discernible improvement in children’s trunk control. In addition, the QUEST and the PIADS questionnaires demonstrated a good acceptability and satisfaction of the garment by the patients and the caregivers.

## 1. Introduction

Cerebral Palsy (CP) is a neurologic condition characterized by non-progressive impairments resulting from a brain injury that takes place during the incomplete development of the cerebral structures [1,2]. This pathology causes symptoms, such as muscle weakness, spasticity, bone deformities, compromised balance and coordination, reduction in walking speed, an increase in the duration of double support, and diminished endurance [1]. These physical limitations can negatively affect the ability to engage in daily activities, integrate into communities, and experience a satisfactory quality of life [1]. The motor impairments of CP patients are usually evaluated by employing validated clinical scales, including (i) the Gross Motor Function Classification System (GMFCS), which focuses on functional abilities related to gross motor skills [3]; (ii) the Modified Ashworth Scale (MAS) [4]; (iii) and the Level of Sitting Scale (LSS), which offers distinct and quantifiable descriptions of various sitting abilities, focusing on the act of sitting as it relates to the activity component of the International Classification of Functioning, Disability, and Health (ICF) [5].

Several pharmacological approaches [6] and rehabilitative techniques [7] have been proposed for the treatment of CP patients, including robotic-assisted gait training [2,8] and transcutaneous electrical nerve stimulation (TENS) [9]. The application of TENS involves the use of surface electrodes to deliver a non-invasive treatment method that can facilitate the enhancement in voluntary motor control in individuals who have sustained injuries to their central nervous system [9]. The intervention has been shown to enhance muscular strength and improve the range of motion of passive joints, while concurrently mitigating pain and spasticity [10]. In this perspective, the Exopulse Mollii Suit (EMS) has been developed by Exoneural Network AB (Danderyd, Sweden) with the aim to foster the self-administration of surface electrical stimulation to the musculature by patients affected by neurological disorders [11]. The tool, shown in Figure 1, is a comprehensive wearable apparatus equipped with integrated electrodes designed to administer electrical stimulation with the objective to mitigate spasticity while concurrently enhancing flexibility and expanding the range of motion.

Specifically, the EMS consists of two main components, namely a wearable suit and a regulating control unit. The portable control unit is responsible for modifying and programming the stimulation according to the specific requirements of the user. The inclusion of menu options that are designed to be easily understood and navigated by the user enables them to exert control over various parameters including intensity, duration, frequency, and stimulation pattern. The suit is equipped with a total of 58 electrodes appropriately positioned across prominent muscle groups. To minimize the occurrence of skin irritation, the electrodes have been specifically engineered to provide a high level of comfort, with a focus on long-term safety and hypoallergenic properties. The electrodes used in the EMS are composed of a conductive substance, such as silicone, in order to enhance the transmission of electrical signals from the control module to the muscular system. The control module transmits electrical impulses to the electrodes based on predetermined configurations. The application of electrical impulses to the electrodes induces a flow of current inside the conductive substance, so activating the muscle. The flow of electrical current is responsible for regulating the processes of muscular contraction and relaxation. Specifically, the electrodes provide varied and subtle impulses to both the contracted and spasmodic muscles, as well as to the feeble muscles that typically oppose them. These impulses facilitate the relaxation of highly tensed muscles by activating their less toned antagonistic muscles, so restoring the natural system that enables them, and consequently the body, to operate in a more coordinated way [12,13]. Indeed, the musculature of the human body functions collectively to sustain equilibrium via the regulation and coordination of reciprocal forces. Spasticity arises from a disruption in the coordination between various muscle groups, resulting in increased muscular stiffness and reduced strength in their opposing counterparts, known as antagonists. This condition ultimately results in significant impairment of motor function. The EMS has been shown to facilitate the restoration of muscle signals disrupted by spasticity and reestablishing the coordinated functioning of muscle units that may have become dysfunctional.

However, it is important to emphasize that the efficacy of the therapy is closely tied to the level of patient compliance, acceptance and involvement in the treatment [14].

It is claimed that “rehabilitation engagement” encompasses various elements, such as the patient’s attitude toward therapy, their level of awareness or recognition of the necessity for treatment, their reliance on verbal or physical cues to engage, their degree of active involvement in therapy exercises, and their consistency in attending the rehabilitation program [15]. Furthermore, various questionnaires have been developed to understand and analyze the user’s satisfaction regarding the technological devices employed in medical therapy. These frameworks encompass factors such as interest, motivation, persistence, and effort, which are considered crucial in the context of rehabilitation therapies. Among these questionnaires, the Quebec User Evaluation of Satisfaction with Assistive Technology (QUEST), a standardized assessment tool for measuring user satisfaction with assistive technology (AT) [16], evaluates the match between an individual’s needs and the features and performance of the AT device they use. In addition, another relevant questionnaire is the Psychosocial Impact of Assistive Devices Scale (PIADS), which is a 26-item self-report questionnaire that assesses the impact of assistive devices on functional independence, well-being, and quality of life [17].

The aim of this study is to evaluate the potential effectiveness of EMS in the clinical treatment of children with CP, focusing on the effect of the therapy on spasticity and trunk control through the administration of GMFCS, MAS, and LSS. Moreover, the compliance of caregivers with respect to the use of the EMS was investigated using the QUEST and the PIADS.

## 2. Materials and Methods

### 2.1. Participants

During the period ranging from 2019 to 2022, a total of 26 children (18 M and 8 F), clinically stable and affected by CP and attending the Neurorehabilitation Unit of the “Padre Pio Foundation and Rehabilitation Centers”, were enrolled in this study. The participants aged from 2 (smallest size of the suit) to 18 years. The exclusion criteria were the presence of magnetically controlled implants (e.g., shunt and baclofen pump) and cardiac pacemaker, heart diseases, malignant courses, infections, fever, skin diseases, and epilepsy. The children were categorized based on the different forms of CP and motor functions, resulting in 22 cases of quadriplegia, 2 cases of diplegia, and 2 cases of hemiplegia. Additionally, the patients were also categorized based on the body involvement, identifying 19 cases as spastic, 5 cases as dyskinetic, and 2 cases as hypotonic. Notably, none of the patients presented ataxia.

A subset of 12 patients bought or rented the device; thus, they received the administration of the EMS-based therapy for one month, whereas the others received only one-hour training. The prolonged duration of the intervention for these 12 patients allowed the administration to them of the questions of the QUEST related to the quality of the service delivery, repair, and follow-up, whereas the other participants did not answer these questions, since they were involved only in 1 stimulation.

The demographic data of the whole study sample and of the subset of 12 participants are reported in Table 1.

### 2.2. Clinical Evaluation and Questionnaire Administration

The children underwent an initial evaluation carried out by skilled clinicians. During this evaluation, various assessment scales were administered to measure their functional abilities. These assessments were conducted at the time of recruitment, which is considered as T0. Specifically, the GMFCS was utilized to evaluate gross motor functions and to classify the study sample [18], while the MAS and the LSS were employed to assess the outcomes of the EMS-based therapy.

In detail, the GMFCS system is based on observational assessment and consists of five levels (I-V) representing different levels of functional limitations, based on a child’s ability to perform gross motor tasks and his/her need for assistive devices or mobility aids. The tool takes into account multiple dimensions, such as mobility, sitting ability, and walking capacity, to capture the child’s overall motor function and functional limitations.

The MAS was employed to assess spastic hypertonicity of the upper limbs (shoulders, elbows, and wrists joints) and lower limbs (hips, knees, and ankles joints). The MAS was devised by Bryan Ashworth, as a means of assessing and categorizing spasticity. The initial iteration of the Ashworth scale consisted of a numerical scale comprising five points, which were utilized to assess the severity of spasticity. This scale ranged from 0 to 4, with a rating of 0 indicating the absence of resistance and a rating of 4 indicating a limb that is rigid either in flexion or extension. Nevertheless, Bohannon and Smith made alterations to the Ashworth scale by incorporating an additional category denoted as 1+ in order to enhance its sensitivity. Therefore, the MAS ranges from 0, indicating no increase in muscle tone, to 4, indicating rigidity in the affected part(s) either in flexion or extension [4].

The LSS was employed for the evaluation of sitting posture. The LSS administration involves a child sitting on a high mat or bench with their thighs supported to the back of the knees and feet unsupported. The sitting position is defined as the child’s hips and trunk that can be flexed sufficiently so that the trunk is inclined at least 60 degrees, and the child’s head position is either neutral with respect to the trunk or flexed. The LSS defines eight levels describing the sitting abilities, relying on the amount of support required to maintain the sitting position and, for those children who can sit independently without support, the stability of the child while sitting [19].

Following the motor function evaluation, participants engaged in a one-hour session while wearing the EMS suit. Importantly, the EMS was set in accordance with specific goals established during the initial evaluation, taking into consideration factors such as targeted muscle groups and desired intensity levels.

The subsequent day, denoted as T1, the children were subjected to identical rating scales.

Moreover, the caregivers were provided with the QUEST questionnaire to assess the users’ satisfaction with the assistive technology, focusing on the first 8 questions (i.e., satisfaction regarding the AT). The QUEST assessment tool is utilized to assess the level of satisfaction among individuals who utilize assistive technology devices, encompassing mobility aids, communication devices, hearing aids, and vision aids. This study centers on various dimensions of assistive technology utilization and its influence on everyday existence. The questionnaire design encompasses various factors, such as those related to devices, services, autonomy, social inclusion, self-esteem, vocational and educational implications, and family repercussions. The utilization of a rating scale, which spans from 1 to 5, is employed for the purpose of evaluating levels of satisfaction [16].

Out of the initial group of 26 children, 12 individuals participated in a training program involving the utilization of the EMS. These participants either purchased or rented the device and followed a specific protocol consisting of three sessions per week for a duration of four weeks. Notably, caregivers were trained by the physician to properly wear the device and to use a preset and customized program according to which each session lasted for 60 min and involved stimulation. Importantly, due to the unique circumstances that prevailed during this study, particularly related to the global health crisis and local regulations, we were unable to gather follow-up clinical measurements of spasticity one month after treatment. The children involved in this study were at home and not in the hospital facilities during this time, which made it challenging to conduct in-person evaluations. However, self-report and caregiver-reported questionnaires were collected, providing valuable insights into the subjective experiences of the children and their caregivers one-month post treatment. Particularly, the psychosocial impact of the device was assessed through the administration of the comprehensive QUEST questionnaire consisting of 12 questions, as well as the utilization of the PIADS. The PIADS is a questionnaire consisting of 26 items that is designed to assess the effects of rehabilitative technologies and assistive devices on the quality of life experienced by users. The instrument is composed of three subscales investigating competence, adaptability, and self-esteem, which are considered essential dimensions encompassed within the construct of quality of life [20]. The competence subscale consists of 12 items that pertain to the perceived functional capability, independence, and performance of individuals. These items include measures of adequacy, efficiency, and skillfulness. The adaptability subscale comprises six items that reflect an individual’s inclination or motivation to engage in social participation and take risks. These items include the ability to actively participate, the willingness to take chances, and the capacity to seize opportunities. The self-esteem subscale consists of eight items that assess various aspects of an individual’s self-confidence, self-esteem, and emotional well-being. These items include measures of one’s sense of control, happiness, and self-confidence [20].

### 2.3. Statistical Analysis

The Shapiro–Wilk normality test was utilized to assess the normality of the distribution of the scores for MAS and LSS. Given the respect of the normality assumption, a paired t-test was conducted to compare the scores across the different sessions (i.e., T0 and T1). The MAS and LSS scores are expressed as mean value and standard deviation. Concerning the satisfaction of the caregivers when handling the EMS, the QUEST scores are expressed as the percentage of the participants who expressed the different grades, whereas the PIADS results are reported as mean value and standard deviation. The statistical analysis was performed using Graphpad Prism software (version 7.0, Graphpad, San Diego, CA, USA). A *p*-value lower than 0.05 was considered statistically significant.

## 3. Results

Following a single session of electrical stimulation with EMS, patients exhibited a statistically significant enhancement in trunk control. Specifically, the LSS increased from 3.84 ± 1.52 to 4.64 ± 1.47 (Figure 2). This improvement was found to be highly significant (t(24) = −9.798, *p* < 0.001). Nevertheless, there was no statistically significant alteration observed in relation to MAS.

The first 8 questions of the QUEST, administered at T1 after one single EMS-based session and after one month, to the participants who underwent the home delivery therapy showed that all the users were satisfied with the dimension, weight, and durability. Moreover, between 80% and 90% of the study sample was satisfied with the safety, ease of use, comfort, adjustment, and after-sales service. Finally, the 76% of the users were satisfied for the effectiveness of the device. These results are summarized in Figure 3A. Figure 3B shows the distribution of the responses to the last four questions of the QUEST administered only to those patients who rented or bought the device, indicative of the psychosocial impact of the equipment. The results showed that 100% of the participants were satisfied regarding the professionality services, and more than the 90% of the users were gratified concerning the service delivery, repair and servicing, and follow-up.

The responses to the PIADS for those who continued the training are summarized in Table 2. The subscale with the best score was adaptability (0.74 ± 0.85), followed by competence (0.67 ± 0.70) and self-esteem (0.59 ± 0.60).

## 4. Discussion

This study investigates the impact of the employment of the EMS on spasticity and the children’s ability to maintain trunk control when affected by CP. Moreover, the users’ satisfaction was investigated by administering the QUEST and PIADS questionnaires.

Specifically, after receiving one session of EMS therapy, LSS showed a discernible improvement in children’s ability to regulate their trunk, whereas the MAS did not show significant changes after one session. The trunk control improvement could be related to the sensory input provided by the EMS-based electrostimulation that can impact proprioceptive awareness, which is essential not only for motor control in dynamic activity but also for maintaining positional control and balance [21,22]. Notably, the electrical stimulation was applied to the antagonists’ muscles in accordance with the reciprocal inhibition mechanism in order to strength the abdominal, dorsal, lumbar, and trunk erector muscles. Specifically, the EMS stimulation is able to trigger inhibitory Ia interneurons in the spinal cord and to reduce the excitability of the agonist’s motor neuron [23,24]. Additionally, other mechanisms of action of the EMS may include neuroplastic changes in brain or spinal cord circuitries [25]. However, a single session could not be sufficient to produce a muscle enforcement able to permanently increase the trunk control. Hence, further studies are indeed necessary to investigate the effect of the EMS-based stimulation on the voluntary motor control and muscles’ strength.

The capacity to accomplish motor actions and skills is inextricably linked to the ability to regulate one’s trunk, making trunk control an essential component of motor development [26,27]. As a result, a segmental evaluation of trunk control needs to be the primary focus area, particularly in children with moderate to severe CP. In this context, the application of the EMS might be focused on the precise segment that is problematic, potentially resulting in a more targeted and accurate intervention that helps further improve segmental control, therefore leading to enhanced performance in functional activities. This finding is in line with a previous study conducted by Raffalt and colleagues [28], demonstrating that 24 weeks of EMS treatment modify the nonlinear dynamics of the trunk accelerations during walking in children afflicted by CP.

Concerning spasticity, it is supposed that a single session of EMS could not act on this symptom, as the neuromodulation effect needs more sessions in this context. Indeed, the process of stimulation can reduce spasticity in muscles by triggering the activation of inhibitory interneurons located in the spinal cord [12]. The reduction in spasticity is believed to occur through the activation of sensory nerve afferents with large diameters, which in turn modulate abnormal interneuron activities across multiple spinal segments. This process is facilitated by an insensitivity to prolonged central excitation resulting from continuous somatosensory stimulation.

With the present study, for the first time ever, the users’ satisfaction was analyzed, proving complete user contentment with the utilization of the device. Particularly, all the PIADS’ subscales showed positive results, implying a beneficial effect of the device on the users. Notably, the self-esteem subscale reported lower positive values, suggesting a reduced improvement in this field. The high score reported by the adaptability subscale reflects the motivation of the individual to participate socially and to adapt to activities of daily living, highlighting a positive attitude of the subject toward the system. The QUEST’s results revealed that participants expressed satisfaction with several aspects of the equipment, including its dimensions, weight, durability, safety, ease of use, comfort, adjustment, and after-sales support. Most respondents, specifically 76%, expressed satisfaction with the effectiveness of the EMS. The results of this study indicate that all participants expressed satisfaction with the level of professionalism exhibited by the service providers. Furthermore, a significant majority, exceeding 90%, reported satisfaction with various aspects of service delivery, including repair, servicing, and follow-up. These results are in line with previous studies investigating the psychophysiological implications of the EMS therapy. For instance, Jonasson et al. [29] conducted interviews with a sample of fifteen patients who had utilized the EMS assistive technology in order to explore and analyze their individual experiences. The participants reported several positive outcomes, which encompassed enhanced mobility, reduced spasticity, and a decrease in the reliance on medication to manage symptoms associated with spasticity. Furthermore, Nordstrom et al. [30] conducted interviews with a sample of six children between the ages of 5 and 10, as well as their respective parents. The interviews were subjected to qualitative content analysis, wherein three primary themes were identified: the impact of the suit on one’s perception, modifications that yield significant outcomes, and strategies for managing the necessity of change. Upon donning the attire, each child documented various effects on their physical well-being, personal identity, and/or specific tasks. In their study, Flodström et al. [12] made an observation regarding the positive impact of the EMS on the level of activity and engagement exhibited by children during self-selected activities. Nevertheless, further investigation involving a greater sample size of young individuals conducted over an extended duration is necessary in order to assess the long-term effects and practicality.

A primary limitation of this study is the limited sample size, specifically in relation to the participants who were able to utilize the EMS over an extended period. Further investigations should strive to integrate a larger number of participants. One additional constraint relates to the absence of a post-intervention period in this study, which would have allowed for an examination of the durability of the enhancements in trunk control. The absence of longitudinal data poses a challenge in determining whether the observed enhancements in trunk control are transient or have an enduring impact on the motor control and overall quality of life of the children. The absence of this aspect underscores the necessity for forthcoming research endeavors to incorporate subsequent evaluations, spanning several months or potentially even years, subsequent to the implementation of interventions.

Another limitation related to the study sample is related to the imbalance between males and females, but this was the ratio M/F we had in our clinic during the enrollement period. It is worth mentioning that gender itself is not a primary determinant of trunk control. In fact, both males and females have similar anatomical structures in the trunk region, including the muscles, bones, and nervous system components responsible for maintaining trunk control. Similarly, the acceptance of the device should not be influenced by gender; however, further studies should indeed investigate this aspect in order to increase the effectiveness and the acceptance of the treatment.

In relation to future investigations, it is recommended that additional research should focus on enhancing the utilization of the EMS across different degrees of severity in individuals with CP. It is imperative to ascertain the potential effects of various application regimes on the efficacy of the intervention. Additionally, it would be valuable to investigate the potential outcomes resulting from the integration of the EMS with complementary therapeutic interventions. In this perspective, future investigations may encompass inquiries that explore the prospective synergistic ramifications stemming from the amalgamation of physical therapy, occupational therapy, or speech therapy in conjunction with the utilization of the EMS. Furthermore, with the continuous progression of technology, it could be necessary to investigate potential avenues for augmenting the design and functionality of the EMS in order to more effectively customize its utilization according to the specific requirements of individual users. Technological advancements encompass the utilization of machine learning algorithms to enhance the customization of electrical stimulation patterns, as well as the integration of biofeedback mechanisms to enable real-time adjustments of suit parameters. In this perspective, it could be fundamental to estimate the parameters of the electrical stimulation across the skin, for instance, evaluating the gradient of the voltage across the biological tissues in order to precisely define the stimulated muscles. In this regard, several modelling approaches of this kind have been proposed for different biological tissues and applications [31]. Concerning the EMS, it should be noted that the distance between the anode and cathode of the stimulation may vary across the muscle districts; hence, consequently, the depth of penetration could be different in the distinct stimulation sites. Indeed, the study of the depth of stimulation and its relationship with the clinical output (e.g., improvement in motor functions, reduction in spasticity) should be investigated in further studies. Finally, it could be fundamental to monitor brain and autonomic nervous system activities during the EMS therapy administration through portable neuroimaging techniques, such as electroencephalography (EEG) and functional near-infrared spectroscopy (fNIRS) [32,33,34,35], and wearables or contactless sensors able to provide information regarding the heart rate variability and the emotional state of the participants [36,37,38,39]. Importantly, monitoring the brain activity across several sessions could provide information about a possible reduction in the excitability of corticomotor neurons and a synaptic reorganization within the sensory and motor cortices [13]. In fact, cerebral reorganization, at the basis of neuroplasticity mechanisms, depends on structural and functional modifications that require longer time intervals than a session. The results of this study indeed highlight the significance of ongoing investigations and advancements in the realm of assistive technology for individuals with CP, fostering the employment of the EMS to improve the trunk control in CP patients, also exploiting the good acceptance of the garment by the users.

## 5. Conclusions

This study investigates the impact of the employment of the EMS on CP children’s ability to maintain trunk control. Specifically, after undergoing a single EMS session, LSS showed a discernible improvement in children’s trunk control. In addition, the QUEST and the PIADS questionnaires demonstrated a good acceptability and satisfaction of the garment by the patients and the caregivers. However, it is important to consider these findings as an initial stage in the progression toward more extensive, varied, and long-term investigations in subsequent research endeavors. In fact, future investigations can offer more definitive proof regarding the usefulness and efficacy of the EMS for pediatric individuals affected by CP.

## Figures and Tables

**Figure 1 brainsci-13-01491-f001:**
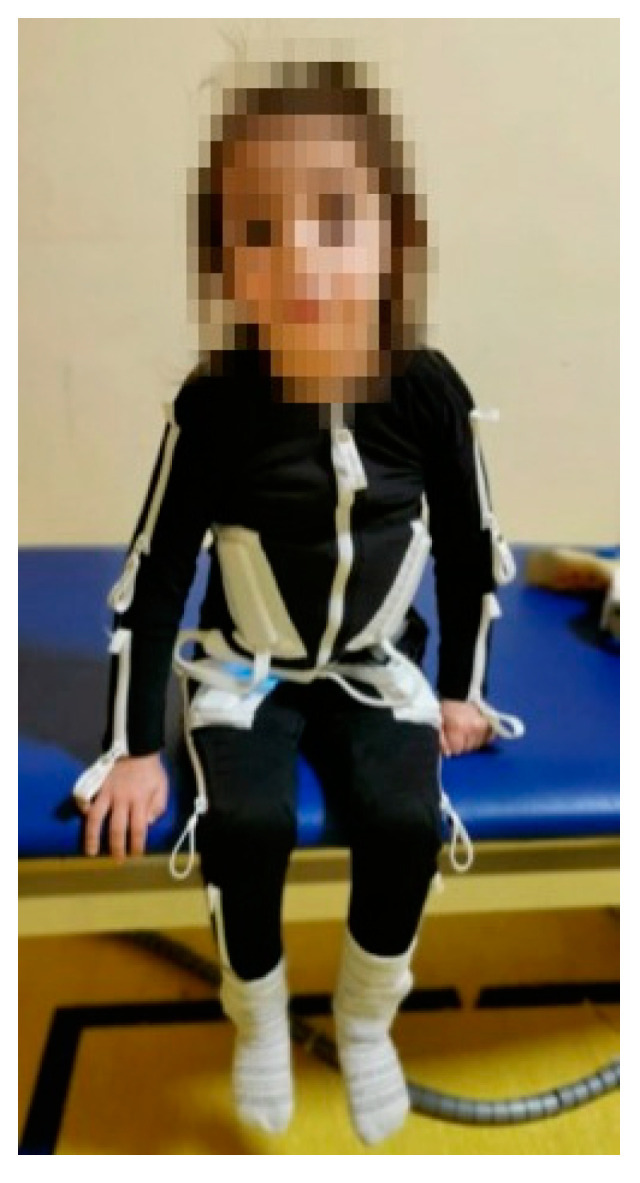
Picture of a representative participant wearing the garment.

**Figure 2 brainsci-13-01491-f002:**
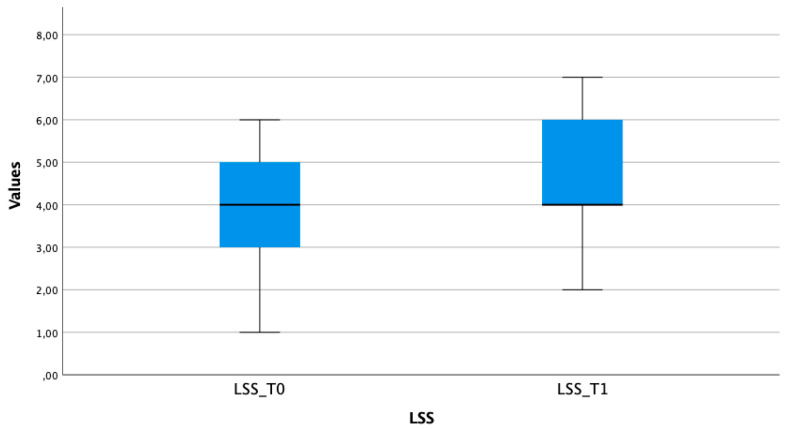
Boxplot reporting the distribution of the LSS scores at T0 and T1. The significant increase in the LSS scores at T1 with respect to T0 highlights an improvement in the trunk control in children with CP after a single EMS-based session.

**Figure 3 brainsci-13-01491-f003:**
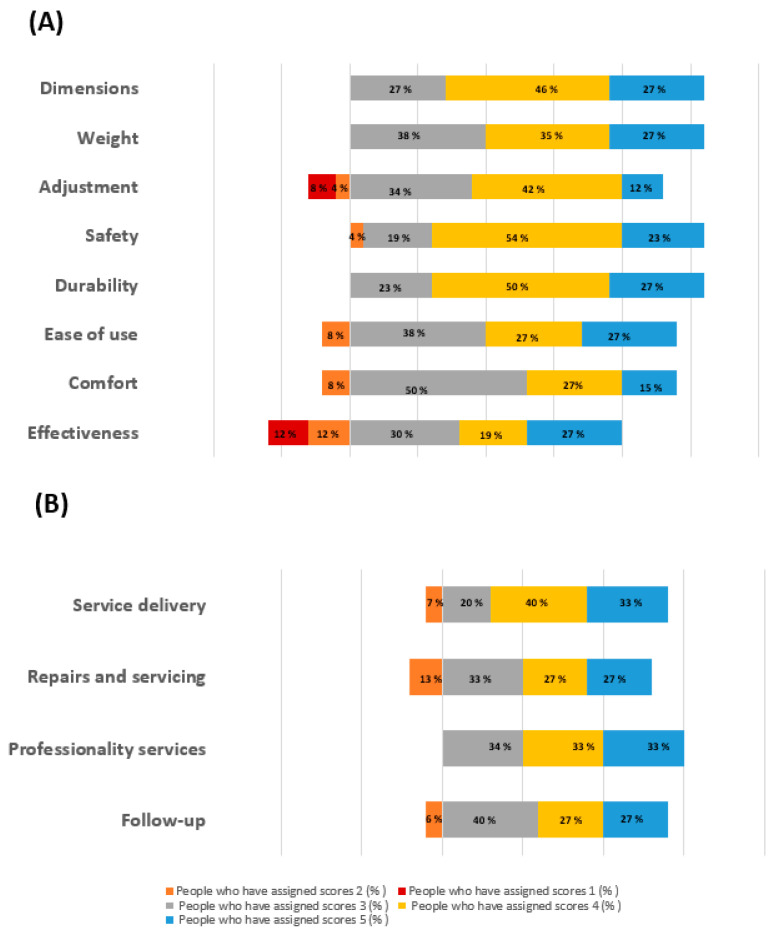
(**A**) Distribution of the participants’ satisfaction regarding the first 8 questions of the QUEST about the features of the AT, and (**B**) distribution of the participants’ satisfaction regarding the last 4 questions of the QUEST about the services related to the EMS assistance. The results are presented as percentage of the total participants.

**Table 1 brainsci-13-01491-t001:** Participants’ demographic data. Notably, the age is expressed as mean ± standard deviation, whereas the qualitative parameters are expressed as absolute frequencies.

Participants	26	12
Age (years)	11 ± 7.00	10 ± 5.69
Gender		
Female	8	2
Male	18	10
Cerebral Palsy		
Quadriplegia	22	10
Diplegia	2	2
Hemiplegia	2	0
Classification		
Spastic	19	8
Spastic-dyskinetic Hypotonic	5	3
	2	1
GMFCS		
I	2	0
II	3	2
III	4	1
IV	9	6
V	8	3

**Table 2 brainsci-13-01491-t002:** PIADS scoring provided by the users who underwent one month of EMS-based therapy.

		PIADS	
	Competence Subscale	Adaptability Subscale	Self-Esteem Subscale
Pt1	1.00	2.33	0.88
Pt2	2.17	1.83	1.50
Pt3	0.33	0.33	0.25
Pt4	0.00	0.00	0.00
Pt5	1.25	1.17	1.38
Pt6	1.17	0.00	0.00
Pt7	0.00	0.00	0.00
Pt8	1.25	1.83	1.50
Pt9	0.00	0.00	0.00
Pt10	0.17	0.67	0.00
Pt11	0.67	0.50	0.75
Pt12	0.08	0.17	0.25
Lowest score	0.00	0.00	0.00
Highest score	2.17	2.33	1.5
Mean	0.67	0.74	0.59
Standard deviation	0.70	0.85	0.60

## Data Availability

Data will be provided on request to the corresponding author.

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
