# Peer review of "Assessing the Impact of Electrosuit Therapy on Cerebral Palsy: A Study on the Users’ Satisfaction and Potential Efficacy"

_brainsci, 2023, doi:10.3390/brainsci13101491_

Round 1

Reviewer 1 Report

The manuscript "Assessing the Impact of Electrosuit Therapy on Cerebral Palsy: A Study on the Users’ Satisfaction and potential efficacy” by Drs. David Perpetuini et al is describing effectiveness of electrosuit therapy in the clinical treatment of children with cerebral palsy. It was reported that the patients exhibited a statistically significant enhancement in trunk control.

 An important topic and interesting, I have no objections on the merits. There are some questions that I would like to clarify.

 Among other things, the authors write that the electrical stimulation was applied to the antagonists’ muscles in accordance with the reciprocal inhibition mechanism in order to strength the abdominal, dorsal, lumbar, and trunk erector muscles. I would like to present this, perhaps a short description of the electrosuite or an explanatory drawing.

What electrodes are used?

Which nerves were most likely affected by stimulation? There are options for calculating the parameters of electrical stimulation through the skin, through the skull and, finally, directly in the living tissue of mammals. This can be easily suggested using a simple calculation algorithm, it was published in many papers, for example:  

Tsytsarev et al, Imaging cortical electrical stimulation in vivo: fast intrinsic optical signal versus voltage-sensitive dyes; Opt Lett . 2008 May 1;33(9):1032-4. doi: 10.1364/ol.33.001032.

The manuscript does not have enough technical information about the electrosuit used - was it the same type used as in the publication below?

Raffalt et al, Electro-suit treatment of children with unilateral cerebral palsy alters nonlinear dynamics of walking Clin Biomech; 2022 Aug;98:105714. doi: 10.1016/j.clinbiomech.2022.105714

If no, it is better to describe the difference at least briefly.

In general results can be interesting for the scientific community.

Author Response

The manuscript "Assessing the Impact of Electrosuit Therapy on Cerebral Palsy: A Study on the Users’ Satisfaction and potential efficacy” by Drs. David Perpetuini et al is describing effectiveness of electrosuit therapy in the clinical treatment of children with cerebral palsy. It was reported that the patients exhibited a statistically significant enhancement in trunk control. An important topic and interesting, I have no objections on the merits. There are some questions that I would like to clarify.

We want to thank the Reviewer for the effort put in revising our manuscript. Below are reported the responses to the Reviewer’s comments and the manuscript was revised accordingly. We do believe that following his advice and comments, the manuscript gained more scientific value.

Among other things, the authors write that the electrical stimulation was applied to the antagonists’ muscles in accordance with the reciprocal inhibition mechanism in order to strength the abdominal, dorsal, lumbar, and trunk erector muscles. I would like to present this, perhaps a short description of the electrosuite or an explanatory drawing. What electrodes are used?

 We thank the Reviewer for this important comment. We modified the Introduction section providing more information regarding the electrodes and the muscles electrical stimulation. Moreover, a picture of the equipment is now reported.

Please refer to lines 73-102:

“Specifically, the EMS consists of two main components, namely a wearable suit and a regulating control unit. The portable control unit is responsible for modifying and programming the stimulation according to the specific requirements of the user. The inclusion of menu options that are designed to be easily understood and navigated by the user enables them to exert control over various parameters including as intensity, duration, frequency, and stimulation pattern. The suite is equipped with a total of 58 electrodes appropriately positioned across prominent muscle groups. To minimize the occurrence of skin irritation, the electrodes have been specifically engineered to provide a high level of comfort, with a focus on long-term safety and hypoallergenic properties. The electrodes used in the EMS are composed of a conductive substance, such as silicone, in order to enhance the transmission of electrical signals from the control module to the muscular system. The control module transmits electrical impulses to the electrodes based on predetermined configurations. The application of electrical impulses to the electrodes induces a flow of current inside the conductive substance, so activating the muscle. The flow of electrical current is responsible for regulating the processes of muscular contraction and relaxation. Specifically, the electrodes provide varied and subtle impulses to both the contracted and spasmodic muscles, as well as to the feeble muscles that typically oppose them. These impulses facilitate the relaxation of highly tensed muscles by activating their less toned antagonistic muscles, so restoring the natural system that enables them, and consequently the body, to operate in a more coordinated way. Indeed, the musculature of the human body functions collectively to sustain equilibrium via the regulation and coordination of reciprocal forces. Spasticity arises from a disruption in the coordination between various muscle groups, resulting to increased muscular stiffness and reduced strength in their opposing counterparts, known as antagonists. This condition ultimately results in significant impairment of motor function. The EMS has been shown to facilitate the restoration of muscle signals disrupted by spasticity and reestablishing the coordinated functioning of muscle units that may have become dysfunctional.”

Which nerves were most likely affected by stimulation? There are options for calculating the parameters of electrical stimulation through the skin, through the skull and, finally, directly in the living tissue of mammals. This can be easily suggested using a simple calculation algorithm, it was published in many papers, for example:  

Tsytsarev et al, Imaging cortical electrical stimulation in vivo: fast intrinsic optical signal versus voltage-sensitive dyes; Opt Lett . 2008 May 1;33(9):1032-4. doi: 10.1364/ol.33.001032.

We want to thank the Reviewer for pointing out this important aspect. The nerves stimulated by the device vary for each subject. In fact, the placement and the stimulation program can be customed for each patient, accordingly to his/her unique needs. However, the EMS is able to activate inhibitory Ia interneurons in the spinal cord and to reduce the excitability of the agonist’s motor neuron. Additionally, other mechanisms of action of the EMS may include neuroplastic changes in brain or spinal cord circuitries.

Estimating the parameters electrical stimulation across the skin could be interesting, however it should be highlighted that the models proposed by the suggested paper is related to the estimation of the gradient of the voltage across the neurons. It could be quite challenging to apply this algorithm as it is to this application because several aspects between the two studies are different (e.g., biological tissue, animals vs. human, method of stimulation). Moreover, it should be noted that the distance between the anode and cathode of the stimulation may vary across the muscle districts, hence the depth of penetration could change across the different districts stimulated. The study of the depth of stimulation and its relationship with the clinical output (e.g., improvement in motor functions, reduction of spasticity) should be investigated in further studies. We modified the discussion section pointing out this crucial aspect.

 Please refer to lines 278-281:

“Specifically, the EMS stimulation is able to trigger inhibitory Ia interneurons in the spinal cord and to reduce the excitability of the agonist’s motor neuron [23,24]. Additionally, other mechanisms of action of the EMS may include neuroplastic changes in brain or spinal cord circuitries [25].”

Please refer to lines 366-375:

“In this perspective, it could be fundamental to estimate the parameters of the electrical stimulation across the skin, for instance evaluating the gradient of the voltage across the biological tissues in order to precisely define the stimulated muscles. In this regard, several modelling approaches of this kind have been proposed for different bi-ological tissues and applications [27]. Concerning the EMS, it should be noted that the distance between the anode and cathode of the stimulation may vary across the muscle districts, hence, consequently, the depth of penetration could be different in the dis-tinct stimulation sites. Indeed, the study of the depth of stimulation and its relationship with the clinical output (e.g., improvement in motor functions, reduction of spas-ticity) should be investigated in further studies.”

The manuscript does not have enough technical information about the electrosuit used - was it the same type used as in the publication below?

Raffalt et al, Electro-suit treatment of children with unilateral cerebral palsy alters nonlinear dynamics of walking Clin Biomech; 2022 Aug;98:105714. doi: 10.1016/j.clinbiomech.2022.105714

 If no, it is better to describe the difference at least briefly.

 Yes, the suit employed in our study is the same of the study by Raffalts et al., now cited in the paper. Some technical information has been added in the Introduction.

Please refer to lines 73-102:

“Specifically, the EMS consists of two main components, namely a wearable suit and a regulating control unit. The portable control unit is responsible for modifying and programming the stimulation according to the specific requirements of the user. The inclusion of menu options that are designed to be easily understood and navigated by the user enables them to exert control over various parameters including as intensity, duration, frequency, and stimulation pattern. The suite is equipped with a total of 58 electrodes appropriately positioned across prominent muscle groups. To minimize the occurrence of skin irritation, the electrodes have been specifically engineered to provide a high level of comfort, with a focus on long-term safety and hypoallergenic properties. The electrodes used in the EMS are composed of a conductive substance, such as silicone, in order to enhance the transmission of electrical signals from the control module to the muscular system. The control module transmits electrical impulses to the electrodes based on predetermined configurations. The application of electrical impulses to the electrodes induces a flow of current inside the conductive substance, so activating the muscle. The flow of electrical current is responsible for regulating the processes of muscular contraction and relaxation. The electrodes provide varied and subtle impulses to both the contracted and spasmodic muscles, as well as to the feeble muscles that typically oppose them. These impulses facilitate the relaxation of highly tensed muscles by preactivating their less toned antagonistic muscles, so restoring the natural system that enables them, and consequently the body, to operate in a more coordinated way. Indeed, the musculature of the human body functions collectively to sustain equilibrium via the regulation and coordination of reciprocal forces. Spasticity arises from a disruption in the coordination between various muscle groups, resulting to increased muscular stiffness and reduced strength in their opposing counterparts, known as antagonists. This condition ultimately results in significant impairment of motor function. The EMS has been shown to facilitate the restoration of muscle signals disrupted by spasticity and reestablishing the coordinated functioning of muscle units that may have become dysfunctional.”

Please refer to lines 292-294:

“This finding is in line with a previous study conducted by Raffalt and colleagues [25], demonstrating that 24 weeks of EMS treatment modify the nonlinear dynamics of the trunk accelerations during walking in children afflicted by CP.”

In general results can be interesting for the scientific community.

We thank the Reviewer for the positive feedback and for the time and effort put in revising our manuscript.

Reviewer 2 Report

This study examined the impact of electrosuit therapy on children with Cerebral Palsy (CP), particularly its effects on spasticity and trunk control. Conducted between 2019 and 2022, it involved 26 children. Twelve of these children underwent a month-long EMS-based therapy, while the rest had a one-hour session. Tools like the Gross Motor Function Classification System, MAS, and LSS were used to evaluate the therapy's effects. Results showed a significant improvement in trunk control after a single EMS session. Most participants expressed satisfaction with various aspects of the treatment, including safety and ease of use. Moreover, the therapy received positive feedback in terms of adaptability, competence, and self-esteem based on the QUEST subscales. 

1. On what criteria were the 12 patients in Table 1 selected?

2. In Table 1, there are more males than females, with a ratio of 18:8. Could this gender disparity influence the results?

Minor editing of English language required

Author Response

This study examined the impact of electrosuit therapy on children with Cerebral Palsy (CP), particularly its effects on spasticity and trunk control. Conducted between 2019 and 2022, it involved 26 children. Twelve of these children underwent a month-long EMS-based therapy, while the rest had a one-hour session. Tools like the Gross Motor Function Classification System, MAS, and LSS were used to evaluate the therapy's effects. Results showed a significant improvement in trunk control after a single EMS session. Most participants expressed satisfaction with various aspects of the treatment, including safety and ease of use. Moreover, the therapy received positive feedback in terms of adaptability, competence, and self-esteem based on the QUEST subscales. 

  1. On what criteria were the 12 patients in Table 1 selected?

We want to thank the Reviewer for this comment. The selection of the 12 participants reported in table 1 is related to the duration of the treatment. In fact, out of the 26 patients enrolled, only 12 rent or bought the device, hence it was possible to administer to them some questions of the QUEST questionnaire related to the service, such as their professionality, and the quality of the service delivery, repair, and follow up. The inclusion criteria of the 12 participants are the same of the whole study sample. This aspect is now better specified in the manuscript.

Please refer to lines 140-143:

“The prolonged duration of the intervention for these 12 patients allowed to administer to them also the questions of the QUEST related to the quality of the service delivery, repair, and follow up, whereas the other participants did not answer to these questions since they were involved only in 1 stimulation.”

  1. In Table 1, there are more males than females, with a ratio of 18:8. Could this gender disparity influence the results?

We thank the Reviewer for this important comment. It is worth to highlight that gender itself is not a primary determinant of trunk control. In fact, both males and females have similar anatomical structures in the trunk region, including the muscles, bones, and nervous system components responsible for maintaining trunk control. Any observed differences in trunk control between genders are likely to be influenced by a combination of factors, including muscle strength, posture, and training, rather than gender alone. However, further studies should indeed investigate this aspect. Concerning the acceptance of the device, it might not be influenced by the gender, however also this aspect should be investigated in further studies. This crucial point is now mentioned in the Discussion section.

Please refer to lines 345-351:

“Another limitation related to the study sample is related to the unbalance between males and females, but this was the M/F ratio we had in our Clinic during the enrollment period. It is worth mentioning that gender itself is not a primary determinant of trunk control. In fact, both males and females have similar anatomical structures in the trunk region, including the muscles, bones, and nervous system components responsible for maintaining trunk control. Similarly, the acceptance of the device should be not influenced by the gender, however, further studies should indeed investigate this aspect in order to increase the effectiveness and the acceptance of the treatment.”